# Patterns of tropical forest understory temperatures

Ali Ismaeel [1], Amos P. K. Tai [1,2], Erone Ghizoni Santos [3], Heveakore Maraia[4,5], Iris Aalto[3,6], Jan Altman [7,8], Jiří Doležal [5,7], Jonas J. Lembrechts [9], José Luís Camargo [10], Juha Aalto [3,11], Kateřina Sam[4,5], Lair Cristina Avelino do Nascimento [12], Martin Kopecký [7,8], Martin Svátek[13], Matheus Henrique Nunes [3,14], Radim Matula[8], Roman Plichta[13], Temesgen Abera [3,15] & Eduardo Eiji Maeda [3,11] ✉

Temperature is a fundamental driver of species distribution and ecosystem functioning. Yet, our knowledge of the microclimatic conditions experienced by organisms inside tropical forests remains limited. This is because ecological studies often rely on coarse-gridded temperature estimates representing the conditions at 2 m height in an open-air environment (i.e., macroclimate). In this study, we present a high-resolution pantropical estimate of near-ground (15 cm above the surface) temperatures inside forests. We quantify diurnal and seasonal variability, thus revealing both spatial and temporal microclimate patterns. We find that on average, understory near-ground temperatures are 1.6 °C cooler than the open-air temperatures. The diurnal temperature range is on average 1.7 °C lower inside the forests, in comparison to open-air conditions. More importantly, we demonstrate a substantial spatial variability in the microclimate characteristics of tropical forests. This variability is regulated by a combination of large-scale climate conditions, vegetation structure and topography, and hence could not be captured by existing macroclimate grids. Our results thus contribute to quantifying the actual thermal ranges experienced by organisms inside tropical forests and provide new insights into how these limits may be affected by climate change and ecosystem disturbances.

Tropical forests host up to half of Earth's biodiversity[1–3]. However, the climatic conditions encountered by organisms in the tropical forests are not yet well understood. Temperature patterns are a fundamental factor defining the survival, growth, and reproduction rate of organisms[4,5], shaping the occupancy, distribution, and diversity of species[6–9]. Therefore, to better comprehend the ecological niches of species, temperature becomes the most important determinant[10–12].

Despite the importance of temperature in ecosystem functioning and services, currently available climate datasets cannot properly capture the range and variability of temperature in tropical forests. The vast majority of products available at regional or global scales provide estimates of open-air temperature, which accurately represent

the conditions over open, well ventilated, homogeneous areas at 2 metres above the ground. These measurements represent the temperature experienced outside the canopy of tropical forests, and they can differ by several degrees from the conditions experienced at understory level below the forest canopy (i.e., the microclimate)[13,14].

Forest structure is a prominent factor driving the fine-scale horizontal and vertical variation in understory temperature[15,16]. During a clear-sky day, most of the incoming shortwave solar radiation is either absorbed or reflected by the forest canopy, which, along with evapotranspiration cooling and damped air mixing, helps to reduce understory temperature[15,17]. At night, on the other hand, forest canopy helps to retain outgoing longwave radiation, leading to a warmer

temperature as compared to open-field conditions[15]. At larger scales, topographic factors such as elevation, slope, and aspect also influence microclimate patterns[18,19]. For example, the pooling of cold air in low-lying terrains, aspect-related exposure to solar radiation, and the temperature lapse rate due to elevation differences are all well documented[20–23].

Although microclimate has been of long-standing interest in ecology, earlier studies had limited scope as they were based on field measurements at single point locations[24–26]. Recent advances in remote sensing, big-data processing, and the growing availability of ready-to-use fine-resolution remote sensing datasets have created a renewed interest in microclimate ecology[9,27,28]. By employing these advanced techniques, ecologists are gaining new insights into the processes underlying microclimate variability at continental scale and postulating the consequences on forest habitats in nexus with changing climate[14,18,29].

In recent years, efforts have been made to reveal the global patterns of understory and near-surface temperatures. The SoilTemp initiative, for instance, pools microclimate data from thousands of temperature sensors spread across the word[30]. Using this database, multiple understory bioclimatic variables have been developed for European Forests at 25 m spatial resolution[14,31]. Similarly, global estimates of soil temperature at a 1 km resolution were also developed based on this database[32]. Nonetheless, this latter study is known to largely extrapolate the estimates in tropical forests, given that the training dataset for the models used in the study were mostly situated in temperate and boreal regions[32]. Furthermore, the spatial resolution of 1 km is also insufficient for many micro-scale ecological studies[33]. Consequently, to date, the spatial and temporal patterns of temperatures inside tropical forests remain unquantified.

Bridging this knowledge gap is fundamental to foster a next generation of ecological and biophysical models in tropical regions, and thus improve our understanding on how living organisms will respond to climate change. In this study, we present fine scale estimates of understory air temperature (i.e., 15 cm above ground) for the global tropical forest. We used a machine learning model trained with in situ temperature data collected between 2016 and 2021 by 180 microclimate sensors spread across three continents (Fig. 1a). The model was driven by satellite observations of forest structural and functional traits, topographic variables, as well as macroclimatic conditions retrieved from atmospheric reanalysis data. We produced 30 m spatial resolution estimates of microclimate temperatures, providing information on both diurnal and seasonal variability, thus providing the understory thermal ranges. Furthermore, we evaluated day-time and night-time temperature offsets (i.e., the difference between macro- and microclimate temperatures), to quantify the capacity of tropical forests to buffer large-scale climate variability. Finally, we demonstrate that our estimates of understory temperature reveal spatial heterogeneity patterns that are otherwise masked in macroclimate datasets.

## Results

We produced pantropic estimates of daily average ($T_{daily}$), day-time ($T_{dt}$) and night-time ($T_{nt}$) understory temperature at 30 m spatial resolution. As a snippet to the modelling results, Fig. 1b–d shows the spatial variation of annual mean $T_{daily}$ in three continents within a $10° \times 10°$ area selected around the equator. Monthly variations of temperatures for each selected area are shown in Fig. 1e–g. For the selected areas, the average $T_{daily}$, $T_{dt}$, and $T_{nt}$ in Central Amazonia were $24.5 \pm 0.5 °C$ (standard deviation over a $10° \times 10°$ area), $26.1 \pm 0.7 °C$, and $23.3 \pm 0.5 °C$, respectively (Fig. 1e). Understory temperatures were slightly cooler in the central areas of the Congo basin compared to Central Amazonia ($T_{daily} = 23.9 \pm 0.6 °C$, $T_{dt} = 25.5 \pm 0.8 °C$, $T_{nt} = 22.6 \pm 0.6 °C$) (Fig. 1f). We observed a higher spatial variability of $T_{daily}$ in the forests of Borneo Island (Fig. 1g), largely due to the strong

topographic heterogeneity; however, temporal variability was low compared to the other two regions ($T_{daily} = 23.9 \pm 1.6 °C$, $T_{dt} = 25.3 \pm 1.7 °C$, and $T_{nt} = 22.9 \pm 1.7 °C$) (Fig. 1g).

The temperature offset ($\Delta T$), which represents the difference between the expected understory temperature and open-air temperature, also showed large spatial and temporal variability (Fig. 2). Southeast Asia showed a relatively stable daily mean $\Delta T$ ($\Delta T_{daily}$) throughout the year, with only subtle seasonal and latitudinal changes. Nonetheless, some areas of Southeast Asia presented exceptionally positive $\Delta T_{daily}$ (i.e., understory temperatures warmer than the macroclimate), where positive $\Delta T_{daily}$ values were observed during the dry season between May and September (Table S1). On the other hand, $\Delta T_{daily}$, as well as the day-time offset ($\Delta T_{dt}$), remained negative throughout the entire year across South America and Central Africa (Figs. 2 and S1). Forests in Africa had some of the highest intra-annual $\Delta T_{daily}$ variability, while lowest values were observed in the forests of Southeast Asia (Table S1).

In South America, areas near the equator displayed little intra-annual fluctuation in $\Delta T$. However, in the southern parts of the Amazon basin, a strong seasonal signal was observed, with $\Delta T_{daily}$ declining during the dry season from July to November. A similar pattern was observed in Africa, with seasonal stable offsets closer to the equator, and amplified seasonal signals at higher latitudes (e.g., beyond 5° S and 5° N). Overall, elevation was an important feature regulating the spatial patterns of $\Delta T_{daily}$ and $\Delta T_{dt}$, with larger offsets (negative values) often observed at higher elevations (Tables S1 and S2).

The night-time offset ($\Delta T_{nt}$) showed a more diverse spatio-temporal pattern primarily driven by elevation and seasonality. In low-elevation forests of South America and Africa near the equator, the $\Delta T_{nt}$ values remained negative during the entire year (Table S3). However, in northern parts of the Amazon Forest (e.g., French Guinea), night-time understory temperatures were warmer during wet seasons (e.g., 0.25 °C warmer between May–Jul and 0.37 °C warmer between Dec–Feb) (Fig. S2g). In Borneo, $\Delta T_{nt}$ was on average 0.87 °C during the wet season (from Nov–Mar) (Fig. S2i). In mid-elevation forests of Eastern Indonesia (above 5° South), $\Delta T_{nt}$ showed a significantly positive signal during the dry season (from Jun–Oct) (Fig. S2l). Across all three continents, intra-annual variability of $\Delta T_{nt}$ was notable in mid- and high-elevation forests located above 5° in both directions of the equator (Table S5).

The spatial patterns of diurnal understory temperature range ($R_T$) during the months of January and August, as well as the monthly values of $R_T$ at selected locations, are presented in Fig. 3. In South America, some forests north of the equator (e.g., in French Guinea) showed a bi-annual seasonal pattern, with the first $R_T$ peak around March, and the second around October (Fig. 3g). These peaks were at the onset of wet seasons. The southern part of the Amazon basin had maximum $R_T$ during the local dry season, between August and September. A similar pattern was observed on the African continent, with higher $R_T$ values during the dry seasons (Fig. 3h, k, Table S6), which alternated between the Northern and Southern hemisphere. In Southeast Asia, forests in north Borneo showed no pronounced $R_T$ peak with only slight fluctuations in various months (Fig. 3i). The forests of Eastern Indonesia showed maximum $R_T$ in November before the start of a rainy season (i.e., from Dec–Mar). Despite the differences in intra-annual patterns, average $R_T$ values in all continents fluctuated between 1.5 and 5 °C (Fig. 3, Table S6), whereas macroclimate $R_T$ ranged between 3 and 7.5 °C.

The spatial heterogeneity of the understory temperatures was assessed using empirical semivariograms fitted with exponential model functions (Fig. 4). The distance at which the semivariograms flatten represents the minimum-distance where observations are no longer spatially-autocorrelated ($d$). The $d$ for understory temperatures ($d_{under}$) was substantially lower than for open-air temperature ($d_{open}$) across all continents, thus providing quantitative evidence that

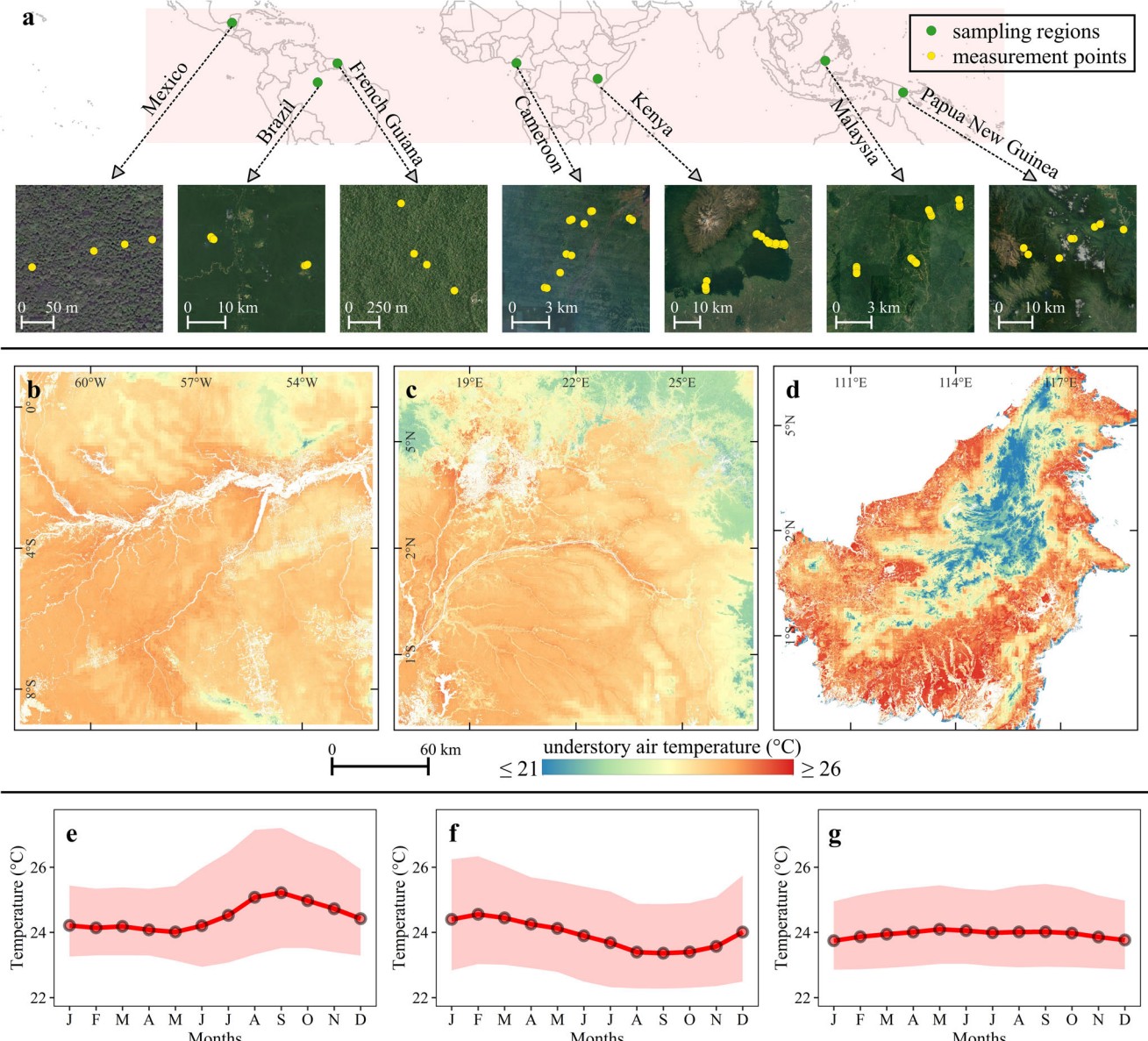

**Fig. 1 | Study area, measurement sites, and modelling output.** Panel **a** depicts the locations of the sampling regions and the distribution of selected measurement points within each region. Panels **b**–**d** illustrates the annual mean spatial variations of modelled understory air temperature within a 10° × 10° area block selected around the equator in South America (**b**), Africa (**c**), and Southeast Asia (**d**). Panels **e**–**g** present the monthly variation of modelled understory air temperature for selected regions of South America (**e**), Africa (**f**), and Southeast Asia (**g**). Each line graph depicts the spatially averaged values for its respective region. The solid red line indicates the mean daily temperature, and the shaded region denotes the range of day-time (upper bound) and night-time (lower bound) air temperatures under the canopy.

microclimate patterns display a higher spatial heterogeneity than what can be inferred from the macroclimate data. These results highlighted that accounting for the effects of vegetation biophysical characteristics and topographic features in regulating temperatures substantially contributes to revealing subtle spatial patterns of thermal traits across tropical forests.

## Discussion

Our study provides the first global estimate of near-ground air temperatures in tropical forest understories, providing a crucial foundation to quantifying the conditions experienced by many organisms in some of the most biodiverse places on Earth. The results reiterate that currently available gridded macroclimate data fail to accurately portray the spatiotemporal patterns and magnitudes of understory temperatures[19,34,35]. We demonstrate that, although the average

understory temperatures in tropical forests are often cooler compared to open-air measurements, the characteristics of these differences vary substantially across different continents, seasons, and time of the day. Temperature offsets, as well as their seasonal fluctuations, were less pronounced near the equator.

At night-time, understory temperatures were often warmer than the macroclimate in some regions. The presence of night-time warming in those regions (Fig. S2) is linked to the shortwave energy absorption by the canopy during the day-time, which is released in the form of longwave radiation at night. Higher heat capacity of forest biomass also helps to dissipate stored energy more slowly making understory warmer at night[17,36,37]. The more energy forests can store within their canopies, the stronger the night-time warming. Furthermore, the retention of surface emitted longwave radiation by forest canopy also contributes to night-time warming[15]. Nevertheless, major

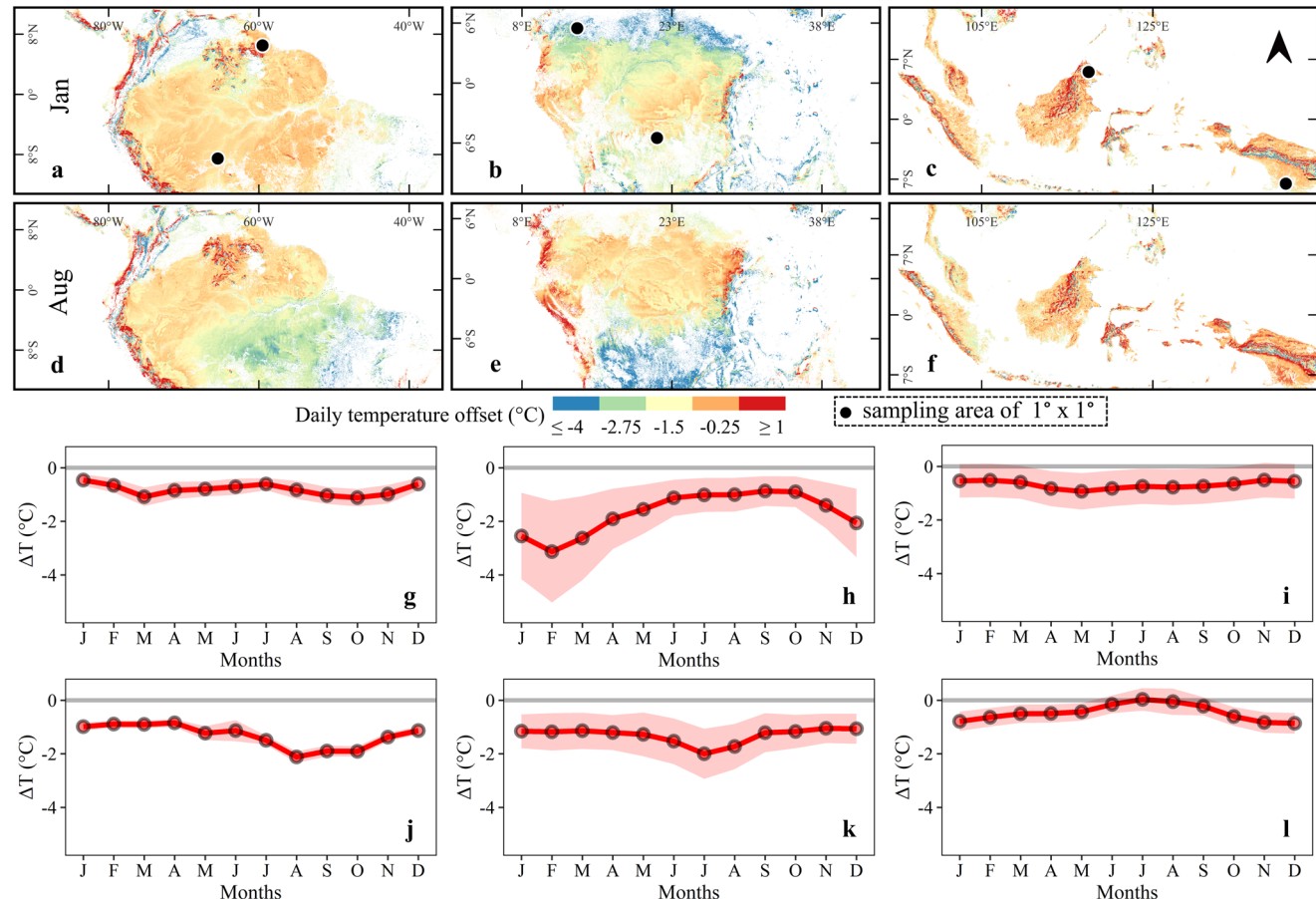

**Fig. 2 | Spatial and temporal variation of mean daily temperature offset.** The offset ($\Delta T_{daily}$) was calculated by subtracting open-air temperature (i.e., ERA5-Land) from modelled understory air temperature. Panels **a**–**f** show the pixel-level variations of $\Delta T_{daily}$ for two months of the year. To present monthly variation, six locations (each of the size 1° × 1° area) were randomly selected on both sides of the equator in South America (**a**), Africa (**b**), and Southeast Asia (**c**). Panels **g**–**l** depict the intra-annual fluctuations of $\Delta T_{daily}$ at the selected locations. Panels **g**–**i** represent monthly $\Delta T_{daily}$ variation for the selected locations in the North while panels **j**–**l** represent $\Delta T_{daily}$ variation in the South. In line graphs, the shaded region around the solid (red) line represents the spatial variation within the selected block of each location.

parts of tropical forest still demonstrate night-time cooling (Table S3) that is in-line with previous research findings[13,17]. Nocturnal transpiration[38] affecting the night-time ambient energy balance[39] is likely the main cause of the observed cooling at night. Studies have reported high nocturnal transpiration rate with increased soil moisture[40,41]. Our results, specially at high latitudes, show profound night-time cooling during wet seasons while understory observe warming during dry season nights (Table S3).

The day-time cooling inside forests can be attributed to the direct effect of biophysical factors on the partitioning of incoming solar radiation between latent and sensible heat[32]. The process of evapo-transpiration (ET) transfers soil water into the atmosphere through the combined effects of plant transpiration and surface water evaporation. The evaporation of soil moisture absorbs the latent heat from the surrounding causing a local cooling effect under the canopy[17,42,43]. Studies have reported a positive relationship between leaf area index (LAI) and ET, ultimately affecting understory cooling[44,45].

Our results showed stronger day-time negative offsets in regions experiencing well-defined dry seasons. These regions are mostly located above 5° in both directions away from the equator (Fig. S1, Table S2), for instance, the southern Amazon basin. Although dry seasons are characterized by lower rainfall water intake, the complex root system of tropical forests can access the deep soil water to maintain ET rates[46–48]. Hence, as macroclimate temperature during dry seasons tends to be higher, the offset in these areas is magnified.

The day-time/daily understory warming in certain regions was also observed during the dry season (Tables S1 and S2). However, these non-intuitive offsets could potentially result from the uncertainties present in the model's input data. For instance, the macroclimate data used in this study is from ERA5-Land, a data source that inherits its own modelling uncertainties. To illustrate the uncertainty tied to ERA5-Land temperature dataset, we compared the monthly temperatures from weather stations with their corresponding ERA5-Land pixel values and reported the correlation and bias for each location (Fig. S12). Although a high correlation exists between weather station data and ERA5-Land data, overall, an underestimation of 1–2 °C is associated with the ERA5-Land temperature data. To overcome this limitation for local applications of the dataset, ground observations from weather stations could be used to bias-correct the open-air temperature from the reanalysis data and thus the temperature offset reported in our study (Fig. S13). Future studies with incorporation of new data from understory loggers installed in diverse conditions will also enable us to overcome the sparse ground data limitation present in this study (Table S5).

Our approach employed remote sensing data in combination with machine learning methods, which allowed us to quantify the importance of biophysical and climatic variables in governing the spatio-temporal behaviours of understory temperatures. Topography (elevation), canopy structure (LAI and Fraction of Absorbed Photo-synthetically Active Radiation (FAPAR)), and open-air temperature

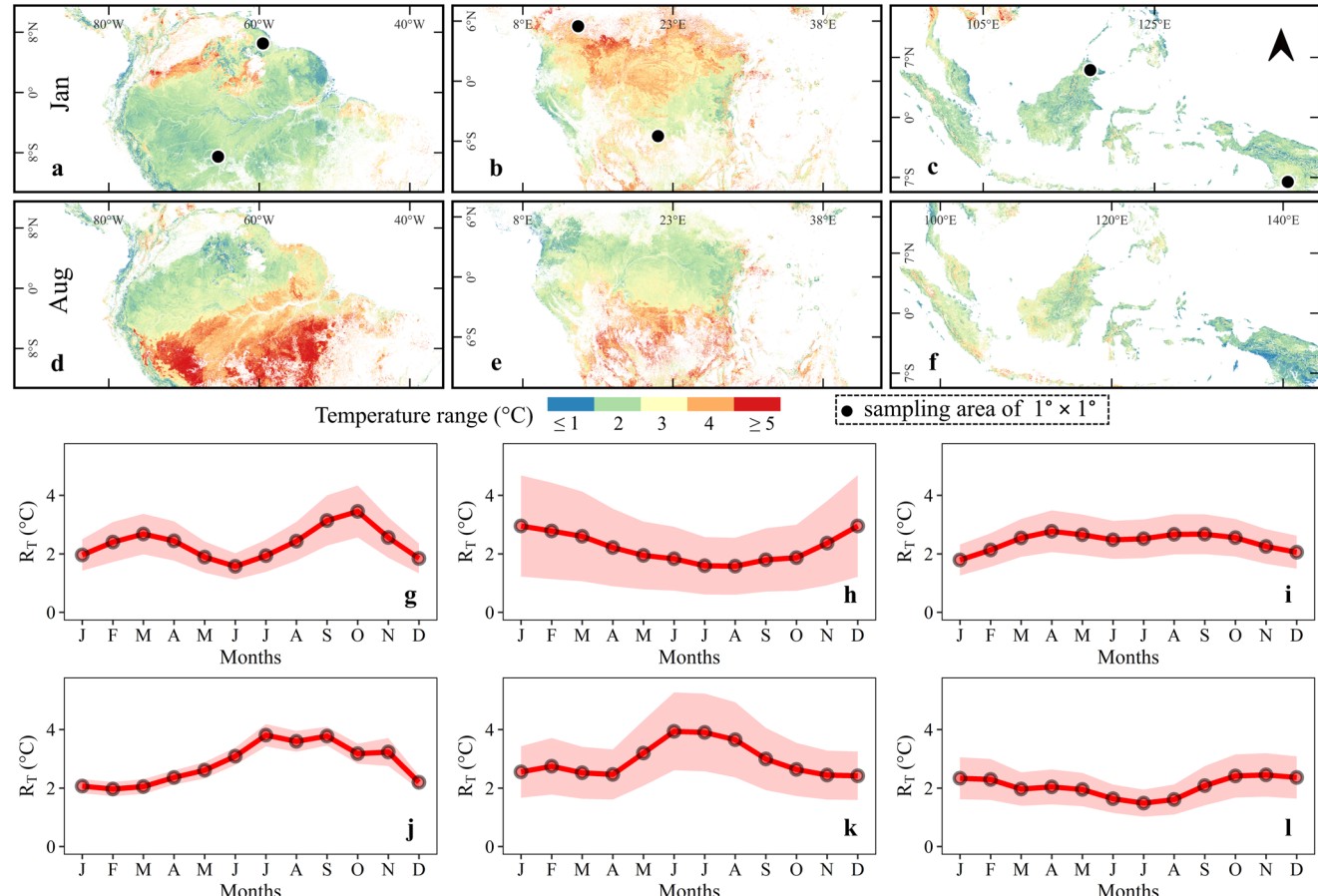

**Fig. 3 | Spatial and temporal variation of diurnal temperature range.** The temperature range ($R_T$) was calculated by subtracting night-time understory air temperature from day-time understory air temperature. Panels **a**–**f** show the pixel-level variations of $R_T$ for two months of the year. To present the monthly variation, six locations (each of the size 1° × 1° area) were randomly selected on both sides of the equator in South America (**a**), Africa (**b**), and Southeast Asia (**c**). Panels **g**–**l** depict the intra-annual fluctuations of $R_T$ at selected locations. Panels **g**–**i** represent monthly $R_T$ variation for the selected locations in the North while panels **j**–**l** represent $R_T$ variation in the South. In line graphs, the shaded region around the solid (red) line represents the spatial variation within the selected block at each location.

emerged as the key climatic and biophysical variables controlling fine-scale variability of understory microclimates across the pan-tropics (Fig. S6). Elevation exhibited a negative relationship, while open-air temperature demonstrated a positive correlation with microclimate. Although it is expected that higher LAI/FAPAR values lead to lower understory temperatures[49], we observed a positive relationship. This partial dependency of the model is merely due to the characteristics of the sample data used for training, which were all located in areas with high vegetation density. The empirical relationship between LAI/FAPAR and microclimate as reported by Hardwick et al.[49], can only be achieved if sensors are installed at larger range of LAI conditions at similar elevations. Nevertheless, the ML approach adopted in the study was able to comprehend the overall importance of canopy structure variables while adapting to with the variables' noncollinearity.

Our model was calibrated under certain biophysical conditions, and predictions outside these conditions are likely to contain higher uncertainties, as machine learning approaches are known to extrapolate estimates outside the boundaries provided during training. To minimize this problem, our training data was gathered considering a large range of geographical settings, across all continents assessed in this study, and different topographical gradients. For instance, our training data from East Africa included microclimate observations collected in tropical forests on Mt Kenya, over an elevational gradient from 1730 to 2450 m a.s.l. We also considered forests under different levels of disturbances, with sensors located in controlled

fragmentation experiments in the Amazon[50,51] and Southeast Asia[52]. Consequently, our estimates by a vast majority (e.g., 83% of the total pixels had a degree of interpolation ≥90%) were within the conditions represented in our training data, with only small areas, mainly high-elevation regions, being extrapolated (Fig. S9). As these uncertainties were quantified and mapped, the degree of interpolation can be used to mask or downweight pixels with larger uncertainties when using the provided maps in ecological applications (Fig. S9).

We have demonstrated through semivariogram analysis that microclimate-informed temperature datasets can unveil spatially independent and heterogeneous habitat conditions. The results of this study provide scientists with more reliable temperature data to support regional, continental, or global assessments in tropical forests. This is a crucial advancement in ecological and global change research as the discrepancies between macroclimate and microclimate temperatures can be substantial in the tropics, leading to biases and erroneous interpretations. For instance, microclimate-informed species distribution models[28,29] have the potential to disclose more robust insights into the various processes underlying species vulnerability to climate change[53]. Climate change exposure can be buffered by microclimate, nonetheless, climate sensitivity can cause microclimate variations impacting the ability of species to cope with it[54]. Furthermore, microclimatic variations affect the spatial patterns of adaptive genetic variation and thus the ability of a population to survive climate change[55,56]. Microclimate also controls the seasonal movements of

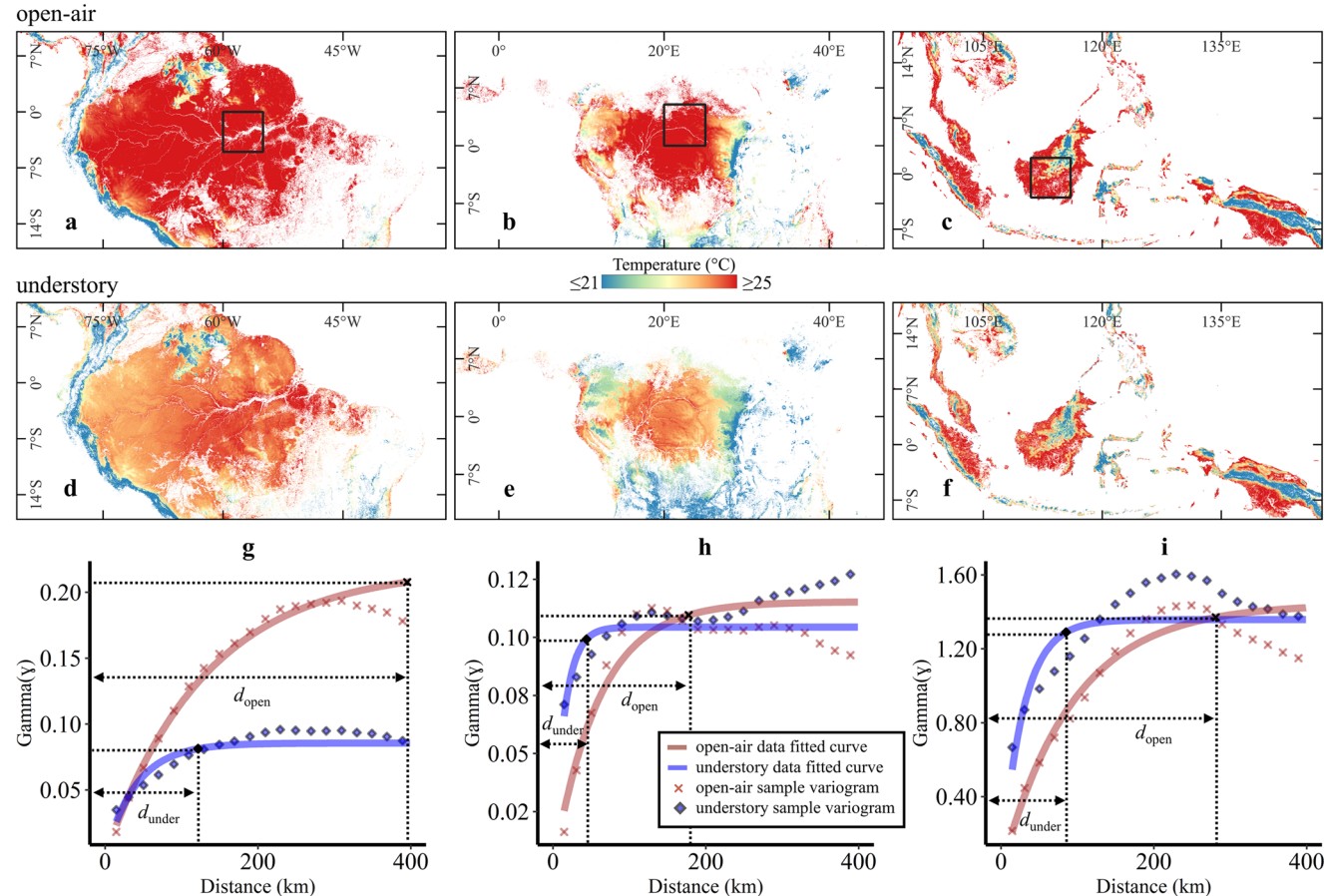

**Fig. 4 | Semivariogram analysis between macroclimate and microclimate datasets.** Panels **a**–**c** and **d**–**f** present the spatial variability of open-air and understory temperatures, respectively. For semivariogram analysis, a 5° × 5° area was selected over three continents (e.g., as depicted in panels **a**–**c**, Central Amazonia in South America, Congo basin in Africa, and Borneo in Southeast Asia).

Panels **g**–**i** depict the semivariogram analysis performed on selected regions of South America (**g**), Africa (**h**), and Southeast Asia (**i**). An exponential model was fitted to the experimental/sample variogram (shown as points in **g**–**i**) values to define sill ($\gamma$) and minimum-distance ($d$) for each dataset.

species within an ecosystem and thus directly impacts distribution capacity and populations especially in fragmented terrains[57,58]. Comprehending how these activities function with microclimate to shape species' cohorts[59] and their exposure to climate change is essential to forecasting range dynamics[60,61].

## Methods

### Study region and temperature data

This study covered the global tropical region between 23° 27′ in the North to 23° 27′ in the South. Within this region, we collected microclimate temperature time-series data with 180 TOMST TMS (Temperature-Moisture-Sensor) dataloggers[62] installed at various locations over the three continents with tropical forests (Fig. 1, Table S5). The temperature data used in this study spans a total of 8 years (i.e., from 2015 to 2022) but the duration of the records differs for each measurement location, ranging from a minimum of 8 months to a maximum of 26 months (Table S5). The TMS loggers are designed to record near-surface soil, surface, and air temperature (°C) every 15 min. In this study, we focused on air temperature measurements, which represent conditions at 15 cm above the ground[62]. The air temperature data from TMS loggers were averaged to hourly dataset in order to be consistent with the temporal resolution of ERA5-Land data. The logger data was converted from UTC to local time zones. The localized hourly mean temperature was then converted to monthly data by averaging (a) 24-h daily temperature ($T_{daily}$), (b) day-time temperature ($T_{dt}$)−temperature records

between 6:00 am to 6:00 pm local time, and (c) night-time temperature ($T_{nt}$)−temperature records between 6:00 pm to 6:00 am local time.

### Explanatory variables

The biophysical variables to include in the modelling were selected based on their relevance to influence forest microclimate based on literature[18,63], spatial resolution, and availability at global scale. In total, 9 biophysical variables (including climatic data) that cover topography, forest phenology and regional macroclimate were used in the study. Topographic layers were derived from a digital elevation model (DEM) of Shuttle Radar Topography Mission (SRTM) at 30 m spatial resolution. Three DEM-based topographic variables, i.e., slope (°), aspect (°) and elevation (m), were used in the model. Forest structural and functional attributes represented by LAI (-), FAPAR (-), and canopy height (CH) (m) were integrated in the model to encompass the forest cover interactions with incoming solar radiation. The LAI and FAPAR data were downloaded from the Copernicus Global Land Service (CGLS) at 300 m, and CH data developed by GLAD (The Global Land Analysis and Discovery) laboratory[64] at a 25 m were used in this study. The LAI and FAPAR data are based on the observations from Sentinel-3 OLCI and PROBA-V satellites[65] whereas CH is based on The Global Ecosystem Dynamics Investigation (GEDI) sensor onboard of the International Space Station. A nearest neighbour interpolation approach was used to harmonize the spatial resolution across different variables.

The hourly gridded data from ERA5-Land reanalysis (from 2000 to 2021) at spatial resolution of 0.1° × 0.1° were used as a macroclimate predictor. Three climatic variables, air temperature at 2 m above the land surface (°C), total precipitation (mm), and surface net solar radiation (J m$^{-2}$) were used as model predictors. Twenty-two years average macroclimate conditions were used in modelling to account for inter-annual variabilities. All the hourly macroclimate variables were in UTC, which were converted to local time using longitudinal information. The local hourly macroclimate variables were then converted to monthly level data as per the average scheme of the microclimate data. Location of microclimate sensors were used to extract the information of biophysical and macroclimate predictors for the training of the machine learning model. The Tree Canopy Cover (TCC) version 4 for the year 2015 at 30 m was used to mask out non-forested area in the region[66]. A threshold of 40% TCC was used for masking the non-forest land. The overall flow diagram of material and method for estimating understory air temperature is shown in Fig. S3.

## Thermal traits: offset, range and spatial heterogeneity

We generated monthly temperature offsets ($\Delta T$) by using microclimate (i.e., understory air temperature at 15 cm above the ground modelled in this study) ($T_{under}$) and macroclimate (i.e., open-air temperature at 2 m above ground provided by ERA5-Land reanalysis) ($T_{open}$) temperature measurements ($\Delta T = T_{under} - T_{open}$) in order to quantify the difference between microclimate and macroclimate across space and seasons. Positive $\Delta T$ values thus indicate warmer forest microclimate conditions, whereas negative values point to a colder forest microclimate. The $\Delta T$ was calculated at daily ($\Delta T_{daily}$), day-time ($\Delta T_{dt}$), and night-time ($\Delta T_{nt}$) level for each month. To better understand the thermal ranges of understory environments, we also estimated the temperature range ($R_T$) using $T_{dt}$ and $T_{nt}$ ($R_T = T_{dt} - T_{nt}$). The analysis of $R_T$ values will help us to better understand the thermal variations between day-time and night-time across space and seasons. For a detailed study of $\Delta T$ and $R_T$ across space and time, we divided $\Delta T$ datasets into three spatial scales based on elevation (i.e., low-elevation (0–500 m a.s.l.), mid-elevation (500–1000), and high-elevation (>1000)) and six scales based on latitude (i.e., low (0°–5° north and south), mid (5°–10° north and south), and high (>10° north and south) latitudes). In total, 18 spatial zones were generated for in-depth analysis of thermal traits (Fig. S14, Table S6).

Finally, a semivariogram analysis was performed on both modelled dataset $T_{under}$ and $T_{open}$ provided by ERA5-Land reanalysis to quantify the spatial heterogeneity of each dataset. For this purpose, a 5° × 5° block near the equator was selected on each continent. The semivariogram analysis was done at a 10 km spatial resolution to keep the pixel size consistent across the datasets. The semivariogram analysis demonstrates the ability to reveal spatially independent thermal conditions by each dataset. As a pre-requisite to semivariogram analysis, detrending of the datasets was carried out by subtracting the best fit surface from the actual data[67]. A linear model was used to define the best fit surface for each dataset using the information of latitude, longitude, and temperature. This process of detrending was instrumental in addressing the dominant physical process that was evident in both datasets and predictably influenced the temperature values[68]. The detrending results of the two datasets (open-air and understory temperature) are shown in Fig. S4. The semivariogram analysis was performed on the residuals of both datasets. Furthermore, the behaviour of experimental/sample variograms were analysed using different combinations of distance and directions (Fig. S5). The semivariogram graph in Fig. 4 is based on 400 km distance in east-west direction (i.e., 90° as shown in Fig. S5). An exponential model curve was used to define the minimum-distance (also called range) and sill parameters which reflect the similarity/heterogeneity of a dataset[69,70]. The point on the fitted curve that corresponds to maximum semivariogram value define the sill (on the y-axis) and minimum-distance (on

the x-axis) for each dataset (Fig. 4). The minimum-distance is a distance at which dataset in question become spatially independent[71].

## Machine learning model

The aim of the study was to maximize the predictive capacity within the biophysical domain covered by the training data. We selected a machine learning (ML) approach as it offers more predictive power compared to other statistical models such as generalized linear models (GLMs) or generalized additive models (GAMs), which are more efficient in exploring predictor inferences[72]. We used a bootstrap aggregating (bagging) regression approach to model understory temperature using the nine macroclimatic and biophysical predictors. The bagging regression model randomly ensembles multiple sets of weak learners and datasets to train the learners in parallel. The model response for new data is generated by aggregating predictions from each weak learner in the ensemble[73]. The bagging algorithm works to minimize the variance and avoid overfitting. It is less prone to outliers and capable to uncover nonlinear/complex relationships of predictors with the response variable (Fig. S6) and can also handle multicollinearity among the predictors (Fig. S7). A 5-fold cross-validation was used to train and test the model performance. Three hyperparameters of ML model, namely (a) minimum leaf size (2–8), (b) number of learners (10–500), and (c) number of predictors to the sample (1–8) were optimized using a grid search approach[74]. A separate ML regression model was ensembled to estimate understory temperature for each temporal scale (i.e., mean monthly daily, mean monthly day-time, and mean monthly night-time).

## Spatial evaluation of model

The ML models are known to be less accurate in extrapolating beyond the boundaries set by the training datasets[75] and should be quantified to indicate model's spatial certainty[76]. Generally, ML models when applied at large spatial scales, are expected to encounter input data that fall beyond the spatial extent encompassed by the training data. In such a situation, a fraction of predictions may fall under the category of the model's extrapolation. To quantify the model spatial certainty, we performed a spatial assessment out-lined by van den Hoogen et al.[76], that quantifies the degree of interpolation and extrapolation at pixel-level. This assessment was done at the monthly level. It helped us to identify the regions that fell outside the bounds of the training data. For model's spatial assessment, monthly-level training data points and the pixels of composite raster were transformed into the same Principal Component (PC) space[76]. Based on our dataset, the first 6 PC axes explained ~93% of the data variation. By combining these 6 PC axes, a total of 15 bivariate spaces were generated; the combinations of these bivariate spaces were as follows: PC1 × PC2, PC1 × PC3, PC1 × PC4, ..., PC5 × PC6. For each of these 15 combinations, every pixel in the composite raster (Fig. S8a) was scored as one if it fell within, or zero if it fell outside the convex hull enclosing the training dataset within that PC combination space (Fig. S8b, c). Pixels falling inside the convex hull were classified as interpolated (Fig. S8d, red points), pixels outside the convex hull were classified as extrapolated (Fig. S8d, blue points). The average of all 15 combinations was taken to quantify the degree of interpolation for each month. At the end, 12 maps of monthly-level spatial extents of interpolation/extrapolation were averaged to present an overall picture of each model's spatial accuracy (Fig. S9).

Finally, to cater the possibility of exaggerated model accuracy because of spatial autocorrelation, we performed a spatial leave-one-out cross-validation analysis to reflect more conservative accuracy parameters for each ML model[76,77]. Under this analysis, a test location was selected, and a buffer zone was established around it. The data points that fall outside of the predefined buffer radius were used to train the model and the test location was used to validate the model prediction. This was repeated for each of the 180 TMS data points. Because of expected spatial autocorrelation close to the validation

point, this process was repeated with an increasing buffer zone around the validation point, each time removing data points that fell within the defined buffer zone from the training data. This method allowed assessing the influence of spatial autocorrelation on the evaluation parameters of each model. The stabilizing of accuracy parameters with increasing buffer radius was an indication of more stable/robust model accuracy indicators (Fig. S10). In addition to above mentioned model evaluation approaches, an independent validation of modelled $T_{daily}$, $T_{dt}$, and $T_{nt}$ was carried out by comparing the results with new independent ground measurements. More details of the data used for blind validation are provided in the supplementary text.

## Data availability
The modelled understory temperatures are made available through the national Finnish Fairdata services (https://www.fairdata.fi/en/). Following link can be used to access the outputs of this study (https://doi.org/10.23729/dd3de08e-39a1-46b0-b28a-7bc577b6c914)[78]. Data in the online repository consists of (a) modelled understory monthly temperatures at 300 m spatial resolution, (b) an index file for 30 m modelled temperatures, and (c) a location file for TMS loggers used in this study. A request for 30 m modelled data using the index file can be made through the corresponding author. The actual logger data can be accessed through SoilTemp platform (https://www.soiltempproject.com/). The details of other model's input variables are mentioned in the 'Methods' section.

## Code availability
Modelling pipeline written in a MATLAB script is available at the following link (https://doi.org/10.23729/dd3de08e-39a1-46b0-b28a-7bc577b6c914)[78]. The script has five parts. The description of each model section is added in the script. Pipeline folder contains test data to understand the modelling flow.

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

## Acknowledgements

This research was partially funded by the Academy of Finland (decision numbers 318252, 319905 and 345472). We are thankful to Thomas Kuiyava from New Guinea Binatang Research Center, who assisted with logistics in Papua New Guinea. K.S. and H.M. acknowledge Starting ERC grant 805189 for funding. J.A., J.D. and M.K. were supported by the Czech Science Foundation (projects

no. 24-11954S and 23-05272S) and the RVO 67985939 project from the Czech Academy of Sciences. M.S. was supported through a grant from the Ministry of Education, Youth and Sports of the Czech Republic (INTER-TRANSFER grant no. LTT19018). R.M. and R.P. were funded by the Ministry of Education, Youth and Sports of the Czech Republic (grant INTER-TRANSFER no. LTT20017). J.J. Lembrechts was supported by the Research Foundation Flanders (FWO, grants 12P1819N, W001919N and 1512720N), as well as by the BiodivERsA-project ASICS (BiodivClim call 2019-2020; G0H6720N). T.A. acknowledges funding from Alexander von Humboldt Foundation.

## Author contributions

E.E.M. and A.I. conceptualized and organized the study. E.E.M. supervised the work. A.I. analysed the data, designed the methodology, performed the modelling, and wrote the codes. A.I. and E.E.M. wrote the manuscript. A.P.K.T., E.G.S., H.M., I.A., J.A., J.D., J.J.L., J.L.C., J.Aa., K.S., L.C.A.N., M.K., M.S., M.H.N., R.M., R.P. and T.A. contributed with data collection from the various locations in the Tropics. All the coauthors contributed to the manuscript revision and drafting rebuttal to the reviewers.

## Competing interests

The authors declare no competing interests.

## Additional information

[1]Earth and Environmental Sciences Programme, Faculty of Science, The Chinese University of Hong Kong, Hong Kong, China. [2]State Key Laboratory of Agrobiotechnology, and Institute of Environment, Energy and Sustainability, The Chinese University of Hong Kong, Hong Kong, China. [3]Department of Geosciences and Geography, University of Helsinki, P.O. Box 68, FI-00014 Helsinki, Finland. [4]Institute of Entomology, Biology Centre of the Czech Academy of Sciences, České Budějovice, Branisovska 31, CZ 370 05, Czech Republic. [5]Faculty of Science, University of South Bohemia, Branisovska 1760, CZ 370 05, České Budějovice, Czechia. [6]School of GeoSciences, University of Edinburgh, Edinburgh EH8 9XP, UK. [7]Institute of Botany of the Czech Academy of Sciences, Zámek 1, CZ-252 43, Průhonice, Czech Republic. [8]Faculty of Forestry and Wood Sciences, University of Life Sciences Prague, Kamýcká 129, CZ-16521, Praha 6-Suchdol, Prague, Czech Republic. [9]Research Group Plants and Ecosystems, University of Antwerp, 2610 Wilrijk, Belgium. [10]Biological Dynamics of Forest Fragment Project (BDFFP) - National Institute of Amazonian Research (INPA), CP 478, 69067-375, Manaus, AM, Brazil. [11]Finnish Meteorological Institute, P.O. Box 503, FI-00101 Helsinki, Finland. [12]Associação SOS Amazônia, Rio Branco, AC 69.905-082, Brazil. [13]Department of Forest Botany, Dendrology and Geobiocoenology, Faculty of Forestry and Wood Technology, Mendel University in Brno, Zemědělská 3, 61300 Brno, Czech Republic. [14]Department of Geographical Sciences, University of Maryland, College Park, MD 20742, USA. [15]Department of Environmental Informatics, Faculty of Geography, Philipps Universität-Marburg, Deutschhausstrasse, 12, 35032 Marburg, Germany. ✉e-mail: eduardo.maeda@helsinki.fi

