## [Peer Review File · Nature Communications]

Patterns of tropical forest understory temperaturesREVIEWER COMMENTS

Reviewer #1 (Remarks to the Author):

This study does two things; first, the authors first create high-resolution (30 x 30 m) maps of tropical forest understory temperatures, based on training data, remote sensing data, and machine learning. Second, the authors analyze their new dataset to explore spatial and temporal patterns in canopy temperature buffering. Every one of us who works in forests, recognizes that temperatures measured by a thermometer in the middle of a field are not representative of what organisms in the forest understory experience. As such, it is hugely valuable to see these analyses and such high spatial resolution, based on models that replicate the training data with high accuracy. As expected, understories were typically cooler during the day, warmer at night, and thus had lower temperature amplitude. However, a fair amount of variability in understory temperature vs macroclimate was also observed. The variability was assigned to things like topography, vegetation structure, and their interactions with macroclimate.

This is a valuable contribution, using an elegant combination of field observations, remote sensing, and machine learning models to create high-resolution temperature predictions in the tropical forest understory, where the vast majority of tropical trees and lianas germinate and seedlings establish, and where a wealth of arthropod, fungus, reptile, and mammal diversity resides at least during a part of its lifecycle. The methods seem appropriate—but I hope a reviewer with more experience in remote sensing and machine learning can confirm this—the conclusions are supported by the data, and the paper is generally well written (but can do with some text editing—see some suggestions in the line comments below).

Some general comments:

- I think that the maps/database of tropical forest understory temperatures is perhaps a more important product of this study than the analysis of general patterns in this dataset, but I don't see any reference to the (current or future) availability of the database. That effectively reduces the product of the study to the analysis part, which would be a pity as people interested in knowing or estimating the temperatures in "their" study forest cannot easily (or at all) replicate the exact analyses.
- The general patterns of forest understory (buffering effect) are well-argued, consistent with published work, and make sense, but the deviations from those patterns deserve some more explanations, even if speculative. Because ultimately, the deviations from the pattern are often more interesting than the pattern itself. E.g., What do we know about the forests that show deviating patterns that could explain the deviation?
- Throughout: Christian Körner has published some papers addressing the distinction between altitude, elevation, and height—because the processes that influence forest understory temperature in mountains are not exclusively the biophysical factors that change with distance from sea level (atmospheric pressure, air temperature), I believe that the recommended term is "elevation", rather than "altitude". It might be worth checking this.

McVicar, T.R. and Körner, C., 2013. On the use of elevation, altitude, and height in the ecological and climatological literature. *Oecologia*, 171, pp.335-337.

Körner, C., 2007. The use of 'altitude' in ecological research. *Trends in ecology & evolution*, 22(11), pp.569-574.

- I did not check this throughout, but I noticed that some of the references that were used to support claims were not the ones that provide original support, but they just make the same claim in their paper. I know this is very common, but it's not correct. In addition, some other references did not say anything about the statement they were supposed to support. I recommend a careful check of what evidence the references provide to support the statements they are used for.

For example:

- o L57. I don't contest the truth of this statement, but the way of using citations is not how it ought to be. Lewis et al. simply state "Tropical forests house over half of Earth's biodiversity" in their abstract and paper as context for their study. They did not do the research to support the statement, so when you make the same statement, this needs to be backed up with references to studies that show that 50% of the earth's biodiversity is in tropical forests, through reviews, meta-analyses, theory, or other research. -Erwin does estimate beetle richness and species richness of other arthropods, but not biodiversity at large.

- o L62. Engelbrecht et al. is a study on drought; the role of temperature is not addressed in the paper

- As I wrote above, the paper is generally well written, but here and there some re-phrasing is needed to avoid confusion, and some text editing is needed. See an incomplete list of examples in the line comments below.

Line comments:

L51. The term "thermal limits" is typically used for physiological limits for organisms. Here it means something else. To avoid confusion, I suggest using another term. Perhaps "thermal ranges", as used in L 115-116.

L104. You can delete "till". "to date" already means "until now"

L125. "shows" instead of "has shown"

L136. Caption can use some text editing. E.g., drop the adjective before "panel" (e.g., L137, L139), and the sentence starting on L139 "...depicts...is shown" needs rephrasing.

L150-151 the elevated "understory" temperature in the dry season, can we assume that the overstory was dry-season deciduous? Are these areas on soils that re-radiate a lot of heat that affects the temperatures close to the soil?

L156. "intra-annual fluctuation in dT"

L174-180. Caption Fig. 2. “calculated by subtracting understory air temperature and open-air temperature”. My understanding from the study is that the open-air temperature is subtracted from the predicted understory temperature to get the offset (positive values indicate warmer understory (e.g., L595-596). The phrasing here in the caption is not clear on this. (also, “line graphs depict”, not “...depicts”)

L184. “north of the equator”, instead of “at the north of equator”

L199. Again, make sure that there can be no doubt about what is subtracted from what here, so I would suggest rephrasing to “calculated by subtracting night-time understory air temperature from day-time understory air temperature”

L247-249. The explanation for nighttime temperatures being higher in the understory makes perfect sense; what explains the observed lower nighttime temperatures?

L268. Photosynthetically Active Radiation, is typically referred to as PAR. Why was FAPAR used? Does this have to do with the method of obtaining the parameter? Some specification might be needed to explain FAPAR.

L272. why would higher PAR lead to lower temperatures? Because direct radiation causes deeper shadow, or because high PAR is associated with higher LAI forest, linking PAR to LAI as effector of lower understory temperatures?

L273-275. isn't the bias in the training data a serious problem? Or does it not matter given the focus on closed-canopy tropical forests/forests with high vegetation density?

L279. I thought non-collinearity is desirable, and it is collinearity of variables that needs dealing with. Am I mistaken, or is this phrased incorrectly here?

L296. “datasets” (plural)

L304-305. See also Vinod et al. (2023) for a review on canopy gradients of thermal sensitivity: Thermal sensitivity across forest vertical profiles: patterns, mechanisms, and ecological implications. *New Phytologist*, 237, 22-47.

L352. “account for”, instead of “cater” (?)

L357. “training of THE machine learning model”

L413. Where does minimum leaf size come from? Is this the plant trait, or is this a machine-learning term? Either way, some explanation to avoid confusion would be useful.

L424, 425, 427. “IN such situation”, “of THE model”, “at THE monthly level”

L462. Rstudio is the user interface. The analyses are done in R (so R, rather than R studio should be acknowledged here)

L463. "whereas" instead of "were as"

Reviewer #2 (Remarks to the Author):

The paper by Ismaeel et al. tries to generate a gridded dataset covering the tropics for understory air temperature at 15 cm above the ground. This height is, albeit to a certain extent arbitrary, chosen since the TMS loggers used for observation report temperatures at this common height. The authors correctly point out that the standard way of reporting air temperature is for open areas, well ventilated, protected from direct sunlight and at 2 m height. Also, most dataproducts available e.g. on spatial grids are referring to this variable for air temperature. In the paper, this is coined the "macroclimate" and is the reference for comparison with the understory temperatures investigated here.

For modelling vegetation activity (e.g. photosynthesis), the understory temperature is obviously the more relevant explanatory variable than the open-air one at 2 m "somewhere else", and the reason for the latter being used so often is simply accessibility of records. To come up with temperature data from below the canopy is very meaningful, and this paper is an ambitious attempt towards that goal.

The observational basis for the understory temperatures is a set of 180 TMS loggers distributed over 7 countries in three continents (The Americas, Africa, and South Asia). Table S5 provides the details. However, it is misleading to state that the data "cover a period of six years (2015-2022 [NB that would be 7 years, or even 8 would they all be full calendar years])" since Table S5 reveals that the actual measurement period is much shorter, down to only 8 months for the 63 sensors in Brazil, and with a maximum of only 26 months. The TMS sensors are logging autonomously with a battery lifetime of approx. 10 years - why are the time series so short? Also, the sensor periods do not overlap, e.g. for the Americas, implying that one has to compare them with macroclimate data from different years for the different regions, a systematic problem if there is significant interannual variability.

From the (satellite?) images provided in Fig. 1 (a), it is impossible to find 180 different locations, e.g. for Brazil it seems to be four rather than 63! Does this indicate that many of the sensors are clumped very close together?

What is the typical (or average) distance between the sensors for a given country? If they are very close, their time series for the 15 cm temperature should be highly correlated, partially devaluating the training data for the RF model.

The original resolution of the time series is 15 minutes, however, the authors never use it, but start by

averaging to 1 hr values. Why is that? Also, whether you first average to 1 hr values and then further to monthly values (l. 325-326), or directly from 15 min to monthly ones can't make any difference since averaging is a linear operation.

The reference ("macroclimate") temperature data are not observations, but rather gridded ERA5 reanalysis data. This is a severe methodological restriction since (i) they come of course with their own biases and uncertainties; often ERA5 data for a given pixel are not strongly correlated to existing station data within the same pixel; and (ii) since the resolution is roughly 10 km, and the TMS loggers are presumably quite close to each other, a set of logger data (which surely will differ) are compared to the exact same reanalysis time series. This implies that small-scale differences due to e.g. canopy cover (LAI, FAPAR) or elevation can't be taken care of.

For the model setup, explanatory variables are considered that are logical candidates for determining temperature, first of all elevation, which also turns out to be the single most important variable for the calculated temperature differences (understory minus macroclimate), and then canopy properties, and other much less important ones. Section 4.2 indicates that this part of the data preparation was probably quite involved, requiring grid harmonization through nearest neighbor interpolation etc. For slope and aspect, the SRTM data are still an excellent data source, but these two turned out to be of very little significance. For LAI and FAPAR, presumably there is a misprint: do you mean Sentinel-2 rather than Sentinel-3?

Taking a total of nine explanatory variables into account, a RF regression model is run and compared to the local (TMS) observations. Here, bagging is used, and the OOB observations are then used for model evaluation. This is a perfectly reasonable method; results show that typical absolute errors are between 0.75 and 1.0 degrees Celsius (l. 478). This is a problem since the target variable Delta T often is also in that range (cf. Fig. 2); it is perfectly possible that even the sign of the difference is wrongly predicted, and part of the non-intuitive results (e.g. that the daily maximum temperature is higher under the canopy than in open spaces) might well be due to this uncertainty.

The paper also puts emphasis on the spatial structure (semivariograms) of open-air versus understory temperatures. The distances where spatial independence is reached (according to the exponential model used, which is a common choice) differ a lot between the two temperature variables. However, the numbers reported are not based on observational networks but on the ERA5 reanalysis data on one hand, which uses kriging for interpolation, and the RF outputs on the other, which uses explanatory variables e.g. related to topography. What we see here (in Fig. 4) is the spatial autocorrelation of the kriging procedure vs. that of the topography, not that of the temperature data per se. The qualitative result (shorter correlations for the understory) is intuitive, but the precise numbers reported (e.g. 44 km or 85 km) are questionable. The authors seem to be aware that they don't quantify the spatial structure of temperature observations directly (l. 216ff), but should be more explicit on the issue and refrain from reporting these results at that precision (like "the understory temperature data in Central Amazonia Forest become spatially independent after 122 km").

In that regard, the detrending performed prior to calculating semivariograms, based on best fit surfaces (l. 385ff and Fig. S4) is not described in sufficient detail. What are the horizontal axes in Fig. S4 (unreadable because of blur)? Why is using the residuals only essential? What type of bias would be

induced when the detrending is skipped?

A positive highlight of the paper is the way it deals with interpolation vs. extrapolation (l. 420f). Where and how much the model is forced to extrapolate compared to the observations, is made rather explicit and precisely quantified based on a thorough PC analysis and 2D-convex hulls for 15 combinations of PC pairs. We have seen many papers where the information about the level of extrapolation is not revealed at all.

It would be tempting to try to verify the model predictions through observations of understory and open-air temperature in close vicinity. They might be scarce and scattered, but it should be possible to get access to 2 m air temperature time series from weather stations close to some of your observations sites? This should be a much more stringent test instead of comparing model output with model output as done in the paper.

Overall, while the data analysis methodology is by and large rather sophisticated and thorough, the observational basis with just a couple of months of records here and there is weak, and the paper tends to overstate its results. The limitations of the approach should be pointed out more clearly, following the comments given here.

Further, there are a non-negligible number of typos, some of them (by no means all) are corrected in the annotated pdf, which also contains further comments and suggestions for improvement. If considered thoroughly, the paper should have publication potential and serves the need for understory temperature data for understanding and modelling vegetation dynamics - not only in the tropics.

Reviewer #1:

This study does two things; first, the authors first create high-resolution (30 x 30 m) maps of tropical forest understory temperatures, based on training data, remote sensing data, and machine learning. Second, the authors analyze their new dataset to explore spatial and temporal patterns in canopy temperature buffering. Every one of us who works in forests, recognizes that temperatures measured by a thermometer in the middle of a field are not representative of what organisms in the forest understory experience. As such, it is hugely valuable to see these analyses and such high spatial resolution, based on models that replicate the training data with high accuracy. As expected, understories were typically cooler during the day, warmer at night, and thus had lower temperature amplitude. However, a fair amount of variability in understory temperature vs macroclimate was also observed. The variability was assigned to things like topography, vegetation structure, and their interactions with macroclimate.

This is a valuable contribution, using an elegant combination of field observations, remote sensing, and machine learning models to create high-resolution temperature predictions in the tropical forest understory, where the vast majority of tropical trees and lianas germinate and seedlings establish, and where a wealth of arthropod, fungus, reptile, and mammal diversity resides at least during a part of its lifecycle. The methods seem appropriate—but I hope a reviewer with more experience in remote sensing and machine learning can confirm this—the conclusions are supported by the data, and the paper is generally well written (but can do with some text editing—see some suggestions in the line comments below).

Reply: We deeply appreciate the reviewer's recognition of our work in modelling understory temperatures in the tropics at high spatial resolution. We are also grateful to the reviewer for the kind and constructive comments, which underscore the significance of our research. Please find our responses to each comment below.

Some general comments:

- I think that the maps/database of tropical forest understory temperatures is perhaps a more important product of this study than the analysis of general patterns in this dataset, but I don't see any reference to the (current or future) availability of the database. That effectively reduces the product of the study to the analysis part, which would be a pity as people interested in knowing or estimating the temperatures in "their" study forest cannot easily (or at all) replicate the exact analyses.

Reply: Our plan is indeed to make the database accessible to the public, allowing other researchers to utilize it for their own analyses. This will be done as soon as the manuscript is accepted, to ensure publishing the latest version of the dataset, with all the reviewers' suggestions already incorporated. Additionally, our modelling pipeline will be disclosed to the public as soon as

this paper is published, enabling future studies to replicate this study and enhance the outcomes by integrating more ground data. In our revised manuscript, we have included information on how users can access the modelled understory data, modelling pipeline, and input data (L533-547, “Data availability” section). In other words, we expect that this study will result not in a static product, but rather in an evolving dataset that will be improved by the microclimate community as more data becomes available in tropical regions.

- The general patterns of forest understory (buffering effect) are well-argued, consistent with published work, and make sense, but the deviations from those patterns deserve some more explanations, even if speculative. Because ultimately, the deviations from the pattern are often more interesting than the pattern itself. E.g., What do we know about the forests that show deviating patterns that could explain the deviation?

Reply: We are grateful for the reviewer’s suggestion to delve deeper into the reasons behind the deviation patterns observed in our study compared to prior published work. This includes phenomena such as certain forest understories experiencing daytime warming during the dry season, and why nighttime cooling is observed in the majority of tropical forests.

The reviewer has generously guided us towards potential explanations. We have updated the manuscript by incorporating these explanations into the discussion section. Please refer to our detailed responses to specific comments below for more information.

- Throughout: Christian Körner has published some papers addressing the distinction between altitude, elevation, and height—because the processes that influence forest understory temperature in mountains are not exclusively the biophysical factors that change with distance from sea level (atmospheric pressure, air temperature), I believe that the recommended term is “elevation”, rather than “altitude”. It might be worth checking this.

McVicar, T.R. and Körner, C., 2013. On the use of elevation, altitude, and height in the ecological and climatological literature. *Oecologia*, 171, pp.335-337.

Körner, C., 2007. The use of ‘altitude’ in ecological research. *Trends in ecology & evolution*, 22(11), pp.569-574.

Reply: While our maps depict the understory air temperature 15 cm above the ground surface, the classification of the study region, as outlined in Table S6, is based on a Digital Elevation Model (DEM). Therefore, we have adopted the suggestion of the reviewer. McVicar & Körner (2013) also defined such scenario under the term elevation.

We have replaced the term “altitude” with “elevation” throughout the manuscript (L161-165, L296, L305, L359, L396).

The terms low-, mid-, and high-altitude were replaced by low-, mid-, and high-elevation (L170, L173, L310, L396-397).

- I did not check this throughout, but I noticed that some of the references that were used to support claims were not the ones that provide original support, but they just make the same claim in their paper. I know this is very common, but it's not correct. In addition, some other references did not say anything about the statement they were supposed to support. I recommend a careful check of what evidence the references provide to support the statements they are used for.

Reply: We have updated the references for several sentences and included the one that offers analytical support. For example, in addition to the ones pointed out by the reviewer, see L64, L77, L83-84 for additional adjustments.

For example:

- o L57. I don't contest the truth of this statement, but the way of using citations is not how it ought to be. Lewis et al. simply state "Tropical forests house over half of Earth's biodiversity" in their abstract and paper as context for their study. They did not do the research to support the statement, so when you make the same statement, this needs to be backed up with references to studies that show that 50% of the earth's biodiversity is in tropical forests, through reviews, meta-analyses, theory, or other research.

-Erwin does estimate beetle richness and species richness of other arthropods, but not biodiversity at large.

Reply: To support the statement "Tropical forests house over half of Earth's biodiversity" we have added following references in the revised manuscript. These studies have quantified the status of various tropical species groups that contribute to global biodiversity e.g., vertebrates, invertebrates, and vascular plants (L57-L58).

Pillay, R., Venter, M., Aragon-Osejo, J., González-del-Pliego, P., Hansen, A. J., Watson, J. E., & Venter, O. (2022). Tropical forests are home to over half of the world's vertebrate species. *Frontiers in Ecology and the Environment*, 20(1), 10-15.

Hamilton, A. J., Basset, Y., Benke, K. K., Grimbacher, P. S., Miller, S. E., Novotný, V., ... & Yen, J. D. (2010). Quantifying uncertainty in estimation of tropical arthropod species richness. *The American Naturalist*, 176(1), 90-95.

Raven, P. H., Gereau, R. E., Phillipson, P. B., Chatelain, C., Jenkins, C. N., & Ulloa Ulloa, C. (2020). The distribution of biodiversity richness in the tropics. *Science Advances*, 6(37), eabc6228.

- o L62. Engelbrecht et al. is a study on drought; the role of temperature is not addressed in the paper

Reply: The above reference was removed from the list and reference of Brown (2014) is added (L64).

- As I wrote above, the paper is generally well written, but here and there some re-phrasing is

needed to avoid confusion, and some text editing is needed. See an incomplete list of examples in the line comments below.

Reply: We have corrected all the errors pointed out by the reviewer. Find below our responses point by point.

Line comments:

L51. The term “thermal limits” is typically used for physiological limits for organisms. Here it means something else. To avoid confusion, I suggest using another term. Perhaps “thermal ranges”, as used in L 115-116.

Reply: The term “thermal limits” throughout the manuscript is replaced by “thermal ranges” to avoid confusion (L51, L392).

L104. You can delete “till”. “to date” already means “until now”

Reply: Corrected (L104).

L125. “shows” instead of “has shown”

Reply: Corrected (L125).

L136. Caption can use some text editing. E.g., drop the adjective before “panel” (e.g., L137, L139), and the sentence starting on L139 “...depicts...is shown” needs rephrasing.

Reply: The caption of Fig. 1 is rephrased to improve clarity and flow. The suggested corrections are also incorporated (L137-145).

L150-151 the elevated “understory” temperature in the dry season, can we assume that the overstory was dry-season deciduous? Are these areas on soils that re-radiate a lot of heat that affects the temperatures close to the soil?

Reply: We find it challenging to assert that the observed deviations are due to leaf abscission in the upper canopy during the dry season. This is because the macroclimate data used for the offset calculation is based on atmospheric reanalysis, which inherently carries uncertainties. In our revised manuscript, we have discussed the biases associated with the explanatory variables (L269-277). This discussion also sheds light on the warmer climate observed under the canopy during the daytime. To underscore the potential bias in the ERA5-Land temperature dataset, we conducted a comparison analysis between temperature records from globally available weather stations and their corresponding ERA5-Land pixel. The outcomes of this comparison are briefly touched upon between lines 276-277, but a detailed discussion can be found under Fig. S12.

L156. “intra-annual fluctuation in dT ”

Reply: Revised as suggested (L157).

L174-180. Caption Fig. 2. “calculated by subtracting understory air temperature and open-air temperature”. My understanding from the study is that the open-air temperature is subtracted from the predicted understory temperature to get the offset (positive values indicate warmer understory (e.g., L595-596). The phrasing here in the caption is not clear on this. (also, “line graphs depict”, not “...depicts”)

Reply: The caption of Fig. 2 is revised in-line with the reviewer’s suggestions to remove ambiguity (L175-178).

L184. “north of the equator”, instead of “at the north of equator”

Reply: Revised (L186).

L199. Again, make sure that there can be no doubt about what is subtracted from what here, so I would suggest rephrasing to “calculated by subtracting night-time understory air temperature from day-time understory air temperature”

Reply: Caption of Fig. 3 is rephrased as suggested to remove confusion (L199-201).

L247-249. The explanation for nighttime temperatures being higher in the understory makes perfect sense; what explains the observed lower nighttime temperatures?

Reply: In the revised manuscript, we have explored the potential cause of night-time cooling. Nocturnal transpiration, as highlighted in previous studies, could be one potential mechanism driving the cooling effect observed under the canopy at night (L248-253).

L268. Photosynthetically Active Radiation, is typically referred to as PAR. Why was FAPAR used? Does this have to do with the method of obtaining the parameter? Some specification might be needed to explain FAPAR.

Reply: The abbreviation FAPAR refers to the Fraction of Absorbed Photosynthetically Active Radiation. We have revised the definition of FAPAR to clarify this (L286-287).

L272. why would higher PAR lead to lower temperatures? Because direct radiation causes deeper

shadow, or because high PAR is associated with higher LAI forest, linking PAR to LAI as effector of lower understory temperatures?

Reply: In the revised manuscript, we have elaborated on the term FAPAR, which is distinct from PAR and signifies the actual proportion of PAR absorbed by the tree canopy. A higher FAPAR implies a greater absorption of incident solar radiation, resulting in a deeper shadow and a reduced understory temperature. Additionally, a higher FAPAR contributes to increased evapotranspiration, leading to a cooling effect related to ET, as previously discussed in the manuscript (L254-261).

L273-275. isn't the bias in the training data a serious problem? Or does it not matter given the focus on closed-canopy tropical forests/forests with high vegetation density?

Reply: Indeed, the reviewer makes a good point. Given that this study concentrates on dense forests where the tree canopy cover exceeds 40% (L380), the bias in the training data does not pose significant concerns.

L279. I thought non-collinearity is desirable, and it is collinearity of variables that needs dealing with. Am I mistaken, or is this phrased incorrectly here?

Reply: Indeed, the presence of non-collinearity among the variables in our model is anticipated. In the revised manuscript, we have rephrased the sentence and replaced the wording “dealing with” with “adapting to” (L297).

L296. “datasets” (plural)

Reply: Corrected (L315)

L304-305. See also Vinod et al. (2023) for a review on canopy gradients of thermal sensitivity: Thermal sensitivity across forest vertical profiles: patterns, mechanisms, and ecological implications. *New Phytologist*, 237, 22-47.

Reply: The suggested reference is added to the sentence (L325)

L352. “account for”, instead of “cater” (?)

Reply: Corrected as suggested (L373)

L357. “training of THE machine learning model”

Reply: Corrected as suggested (L378)

L413. Where does minimum leaf size come from? Is this the plant trait, or is this a machine-learning term? Either way, some explanation to avoid confusion would be useful.

Reply: Leaf size is a hyperparameter of machine learning model. A clarification is added in the sentence to avoid confusion (L438).

L424, 425, 427. "IN such situation", "of THE model", "at THE monthly level"

Reply: All the suggestions are incorporated, and sentences are revised (L449, L450, L452).

L462. Rstudio is the user interface. The analyses are done in R (so R, rather than R studio should be acknowledged here)

Reply: Corrected as suggested (L485-486)

L463. "whereas" instead of "were as"

Reply: Corrected (L486)

Reviewer #2:

The paper by Ismaeel et al. tries to generate a gridded dataset covering the tropics for understory air temperature at 15 cm above the ground. This height is, albeit to a certain extent arbitrary, chosen since the TMS loggers used for observation report temperatures at this common height. The authors correctly point out that the standard way of reporting air temperature is for open areas, well ventilated, protected from direct sunlight and at 2 m height. Also, most dataproducts available e.g. on spatial grids are referring to this variable for air temperature. In the paper, this is coined the "macroclimate" and is the reference for comparison with the understory temperatures investigated here.

For modelling vegetation activity (e.g. photosynthesis), the understory temperature is obviously the more relevant explanatory variable than the open-air one at 2 m "somewhere else", and the reason for the latter being used so often is simply accessibility of records. To come up with temperature data from below the canopy is very meaningful, and this paper is an ambitious attempt towards that goal.

Reply: We express our gratitude to the reviewer for emphasizing the significance of incorporating an understory temperature profile in tropical forests for enhancing the accuracy of future modeling efforts across ecology. Please find our responses to each comment below.

The observational basis for the understory temperatures is a set of 180 TMS loggers distributed over 7 countries in three continents (The Americas, Africa, and South Asia). Table S5 provides the details. However, it is misleading to state that the data "cover a period of six years (2015-2022 [NB that would be 7 years, or even 8 would they all be full calendar years])" since Table S5 reveals that the actual measurement period is much shorter, down to only 8 months for the 63 sensors in Brazil, and with a maximum of only 26 months. The TMS sensors are logging autonomously with a battery lifetime of approx. 10 years - why are the time series so short? Also, the sensor periods do not overlap, e.g. for the Americas, implying that one has to compare them with macroclimate data from different years for the different regions, a systematic problem if there is significant interannual variability.

Reply: We have revised the sentence where ground observation data is mentioned to avoid any confusion. In the revised manuscript, we now mention that the temperature data utilized in this study spans a total of eight years (i.e., from 2015-2022). However, the duration of the records differs for each measurement location, ranging from a minimum of eight months to a maximum of 26 months (L339-342).

While sensor batteries are designed to last approximately 10 years, their operational lifespan in forest environments is often shortened by falling tree debris and disturbances caused by ground-dwelling animals. These sensors are susceptible to their installation conditions, and the factors mentioned earlier can introduce discrepancies in the temperature readings. For this study, we

have used temperature records within a stable installation environment. Furthermore, in some cases (e.g., in Mexico), sensors were often installed as part of (ecological) research with their own - other - goals, and using funding for limited time periods only, and that thus sensors were sometimes removed after a fixed period. Other sensors have only been installed recently and more recent data is thus not yet available. For this study we have employed the best available data resources. In case of Brazil, only 8 month of data was available at the time of this study.

To overcome the interannual variability challenge for macroclimate data, monthly average data for twenty-two years (2000-2021) were used in this study as previously mentioned in L372-373. With additional analysis in the revised manuscript, we have provided evidence that even with a small observation period of 8 months, our RF model is able to capture the seasonal variations at those locations (Fig. S13 (c)).

Nevertheless, since our study represents a crucial first step towards understory mapping in the tropics, there is ample scope for improvement. By integrating more ground observations in future studies, the accuracy of understory temperature mapping can be significantly improved. This is an important point we must clarify. We expect that this study will not result in a static product, but rather in an evolving dataset that will be improved by the microclimate community as more data becomes available in tropical regions. To facilitate this, our modelling pipeline will be disclosed to the public, enabling future studies to replicate and enhance this study's limitations by integrating more ground data. In our revised manuscript, we have included information on how users can access the modelled understory data, modelling pipeline, and input data (L533-547, "Data availability" section).

From the (satellite?) images provided in Fig. 1 (a), it is impossible to find 180 different locations, e.g. for Brazil it seems to be four rather than 63! Does this indicate that many of the sensors are clumped very close together?

What is the typical (or average) distance between the sensors for a given country? If they are very close, their time series for the 15 cm temperature should be highly correlated, partially devaluating the training data for the RF model.

Reply: The satellite images shown in Fig. 1 (a) do not show all the sensors. We have revised the Fig. 1 caption and indicated that the distribution of sensors in each country is only for selected points (L137-138). The information of all TMS logger locations is shared as point file over the online repository where other data is shared with the public (L535-L538). Table S5 is revised and information of average distance between sensors at each location is added. The distance between sensors varied broadly, as some countries contained multiple study sites (with distances between study sites varying from 1 to 130km), and many sensors could be located inside each study site (with distance between sensors varying from 10 to 1000m).

Keeping in mind the high spatial correlation in some of the training datasets, we explicitly selected a bagging ensemble approach for RF algorithm (as already mentioned in L429-433). The bagging is a “**parallel**” ensemble method and is different from other ensemble techniques like boosting (e.g., gradient, or adaptive). In boosting approach, decision trees are trained sequentially using weights for the prediction and every next decision tree learns from previous decision tree predictions. On the other hand, bagging does not assign weights to predictions. In bagging ensemble, each decision tree is developed independently in parallel using a bootstrapped sampling of the original dataset, and each split in each tree is based on a random subset of input data. However, if a fraction of training dataset is highly correlated, it might end up being selected more often across different trees. This doesn’t necessarily mean that the algorithm is biased towards these training datasets. It’s worth noting that the power of a bagging-based RF model comes from aggregating predictions across many trees, each of which might have used a different subset of training data. This can help to balance out the influence of any correlated training data and ensure that the final prediction is based on a broad range of information.

Furthermore, to ensure that some degree of spatial-autocorrelation would not cause an overestimation in the model accuracy, we also conducted a spatial leave-one-out cross-validation analysis (Fig. S10), as suggested in Ploton et al., (2020).

Ploton, P., Mortier, F., Réjou-Méchain, M., Barbier, N., Picard, N., Rossi, V., ... & Pélissier, R. (2020). Spatial validation reveals poor predictive performance of large-scale ecological mapping models. *Nature communications*, 11(1), 4540.

The original resolution of the time series is 15 minutes, however, the authors never use it, but start by averaging to 1 hr values. Why is that? Also, whether you first average to 1 hr values and then further to monthly values (l. 325-326), or directly from 15 min to monthly ones can't make any difference since averaging is a linear operation.

Reply: Indeed, averaging is a linear operation, and 15-minute resolution data can be directly converted to monthly scale. Nevertheless, we averaged the TMS observations to an hourly resolution in order to be consistent with ERA5-Land data, which is at this resolution as well. We now mention this fact on line 345-346.

The reference ("macroclimate") temperature data are not observations, but rather gridded ERA5 reanalysis data. This is a severe methodological restriction since (i) they come of course with their own biases and uncertainties; often ERA5 data for a given pixel are not strongly correlated to existing station data within the same pixel; and (ii) since the resolution is roughly 10 km, and the TMS loggers are presumably quite close to each other, a set of logger data (which surely will differ) are compared to the exact same reanalysis time series. This implies that small-scale differences due to e.g. canopy cover (LAI, FAPAR) or elevation can't be taken care of.

Reply: We agree with the reviewer that there is a significant methodological restriction coming with the use of ERA5 reanalysis data. However, multiple recent studies across different regions have reported a high correlation between ERA5-Land temperatures and ground observations from weather stations (Vanella et al., 2022; Zhao et al., 2022; Zou et al., 2022). This high correlation enables the reanalysis data to accurately capture the seasonal temperature variations. However, these studies have reported various magnitudes of the biases for their analysis. The value of bias can vary for each reanalysis pixel and depend on the reference data. We could not find any published study that has reported the correlation and biases for the tropical forests when comparing ERA5-Land temperatures with weather station observations.

However, to directly inform the readers about the accuracy of ERA5-Land data in the tropics, we now conducted a comparison analysis for average monthly temperatures and included the results in the supplementary materials of the revised manuscript (Fig. S12).

For this purpose, we selected active weather stations from the Global Historical Climatology Network – Daily (GHCN-Daily) (Menne et al., 2012) and extracted the data from within our study period (2000-2021). The weather station data were averaged to monthly scale and compared with corresponding ERA5-Land pixel temperatures from the same location. A total of 432 weather stations from across the tropics were used in this comparison. Fig. S12a (see below) shows the spatial distribution of points where correlation (Pearson (r)) values were statistically significant ($p < 0.05$). Only 10 reanalysis pixels showed $r \leq 0.5$. Moreover, for each location, pixel-wise biases were calculated, and the spatial variability is shown in Fig. S12b. Overall, ERA5-Land temperature is underestimated by 1-2 °C. The mean bias was calculated using this formula: $bias = \frac{1}{n} \sum_{i=1}^n (T_{ERA_i} - T_{WS_i})$ where n is total observations of ERA5-Land (T_{ERA}) and weather station temperature (T_{WS}).

This inherited underestimation of ERA5-Land data is likely to affect temperature offset magnitudes that we have reported with reference to ERA5-Land temperature data. We now discuss this limitation in the revised manuscript (L270-277) (for more details on how to overcome this limitation by using local weather station data, please refer to our response to the relevant comment below).

It's important to remember that weather station data has its own set of constraints. Even if we assume that weather station data is free from systematic and reporting errors (which likely is not the case for GHCN-Daily weather data as reported by Menne et al., (2012)), a weather station only records the weather conditions of a specific environment where it is installed. The stations used in our comparative analysis are mostly situated in urban areas. On the other hand, the ERA5-Land pixel depicts the average climate conditions across a 10×10 km area.

The semivariogram analysis as reported in Fig. 4 and on lines 209-217 specifically addresses the reviewer's concern regarding the ability of the RF model to incorporate the small-scale differences in canopy cover and terrain into the coarse macroclimate pixel. The results from this analysis indicate that there is a higher degree of spatial heterogeneity in the modelled understory temperature compared to the macroclimate data (L209-214). This spatial variation was captured through the installed TMS locations that cover a variety of independent biophysical conditions as

shown in Fig. S6. Note also that it is the specific goal of the study to modify/correct the coarse resolution macroclimate temperature data with reference to actual observations of understory temperature by integrating different biophysical variables (e.g., LAI, FAPAR, and DEM).

Fig. S12: Comparison of weather station monthly temperature observations with corresponding ERA5-Land pixel, including the Pearson correlation (top) and the temperature difference between the two (bottom). Negative temperature biases indicate that ERA5-Land is predicting lower temperatures than is observed within weather stations.

References for this reply

- Zhao, P., & He, Z. (2022). A first evaluation of ERA5-Land reanalysis temperature product over the Chinese Qilian Mountains. *Frontiers in Earth Science*, 10, 907730.
- Vanella, D., Longo-Minnolo, G., Belfiore, O. R., Ramírez-Cuesta, J. M., Pappalardo, S., Consoli, S., ... & Gandolfi, C. (2022). Comparing the use of ERA5 reanalysis dataset and ground-based agrometeorological data under different climates and topography in Italy. *Journal of Hydrology: Regional Studies*, 42, 101182.
- Zou, J., Lu, N., Jiang, H., Qin, J., Yao, L., Xin, Y., & Su, F. (2022). Performance of air temperature from ERA5-Land reanalysis in coastal urban agglomeration of Southeast China. *Science of The Total Environment*, 828, 154459.

Menne, M. J., Durre, I., Korzeniewski, B., McNeal, S., Thomas, K., Yin, X., ... & Houston, T. G. (2012). Global historical climatology network-daily (GHCN-Daily), Version 3. NOAA National Climatic Data Center, 10(10.7289), V5D21VHZ.

For the model setup, explanatory variables are considered that are logical candidates for determining temperature, first of all elevation, which also turns out to be the single most important variable for the calculated temperature differences (understory minus macroclimate), and then canopy properties, and other much less important ones. Section 4.2 indicates that this part of the data preparation was probably quite involved, requiring grid harmonization through nearest neighbor interpolation etc. For slope and aspect, the SRTM data are still an excellent data source, but these two turned out to be of very little significance. For LAI and FAPAR, presumably there is a misprint: do you mean Sentinel-2 rather than Sentinel-3?

Reply: We agree with the reviewer's argument that slope and aspect are significant terrain features that influence microclimate conditions. For instance, sun angle variation during the day makes the aspect an important factor to regulate microclimate. Nonetheless, in our case, we averaged the daytime conditions between 6am-6pm (L347-350), which likely contributed to reducing the influence of aspect in mapping understory temperature conditions.

The LAI and FAPAR products are based on Sentinel-3 as written, we have now added the instrument name in addition to the satellite name e.g., Sentinel-3-OLCI (CGLS, 2022) in the revised manuscript to clarify this (L365-366).

CGLS, (2022). Copernicus Global Land Service. Providing Bio-Geophysical Products of Global Land Surface. Available online: <https://land.copernicus.eu/global/themes/vegetation> (accessed in 2022).

Taking a total of nine explanatory variables into account, a RF regression model is run and compared to the local (TMS) observations. Here, bagging is used, and the OOB observations are then used for model evaluation. This is a perfectly reasonable method; results show that typical absolute errors are between 0.75 and 1.0 degrees Celsius (l. 478). This is a problem since the target variable Delta T often is also in that range (cf. Fig. 2); it is perfectly possible that even the sign of the difference is wrongly predicted, and part of the non-intuitive results (e.g. that the daily maximum temperature is higher under the canopy than in open spaces) might well be due to this uncertainty.

Reply: We thank the reviewer for the insightful **comment**. Firstly, we must clarify that the target variable of our RF model was understory temperature, not temperature offset. The range of absolute error shown in Fig. S10 thus represents the model's uncertainty in predicting "understory temperature," not the variable delta T (i.e., offset temperature).

Secondly, to minimize the model's uncertainty, we employed a comprehensive parameterization approach (i.e., a grid search, as mentioned on L437-440). **Nevertheless**, given the vast stretches of the tropical forests and the limited ground data available, relatively high absolute errors were anticipated. This issue can be addressed in future studies by incorporating more ground data.

The concern of the reviewer is valid regarding ΔT magnitude and its sign because ΔT is calculated with reference to ERA5-Land temperature that has its own biases as we have also shown in Fig. S12b. The non-intuitive results (e.g., understory warming during day-time) could indeed potentially result from the uncertainties associated with ERA5-Land data. However, given the overall underestimation in ERA5-Land data as reported in Fig. S12, and the consistency of results (96% of monthly day-time offsets below 0°C) suggests that the general direction of our findings is unlikely to be erroneous. In the revised manuscript, we have therefore now discussed this study limitation. Additionally, we now provide an example to overcome this limitation by employing localized weather station conditions to correct the bias in the ERA5-Land data and the ΔT magnitude reported in our study (L277-280, Fig. S13). We have employed a simple delta change approach (Ho et al., 2012; Hay et al., 2000) to bias-correct the ERA5-Land data and the temperature offset magnitudes. More detailed monthly-level correction using other approaches like multiple linear regression or quantile mapping approach can also be adopted.

To illustrate the example, we have used seven weather stations data from all those countries where TMS loggers were installed (Fig. S13a). Monthly average temperatures (Menne et al., 2018) were used to compare with corresponding ERA5-Land pixel and the mean bias (change factor) was calculated (Fig. S13b). The biased ERA5-Land temperature was adjusted using the calculated mean bias to estimate the bias-corrected reanalyzed temperatures (Fig. S13c). For example, the ERA5-Land pixel values in Mexico (MX) (Fig. S12b) were on average 1.07 (°C) lower when compared with weather station data. Therefore, 1.07 was added to the uncorrected ERA5-Land pixel values (as shown in red color line Fig. S13c) to get bias-corrected ERA5-Land temperature (i.e., dark-green line, Fig. S13c). This corrected ERA5-Land temperature was then used to calculate actual temperature offsets as shown in Fig. S13d. The understory temperature shown in Fig. S13c is from a nearby forest area.

The estimated mean bias can also be directly added to the uncorrected temperature offsets (i.e., reported in this study) to derive bias-corrected temperature offsets. For example, estimated mean bias for MX is -1.07. This bias can be directly added to the uncorrected temperature offsets (orange line, Fig. S13d) to get the bias-corrected temperature offsets (light-green line, Fig. S13d). Given our limited access to worldwide weather station data, especially the lack of access to average day-time and night-time temperature conditions, the end-users of the provided understory temperatures can perform local bias-correction using their national or regional weather station archives.

References for this reply

Ho, C. K., Stephenson, D. B., Collins, M., Ferro, C. A., & Brown, S. J. (2012). Calibration strategies: a source of additional uncertainty in climate change projections. *Bulletin of the American Meteorological Society*, 93(1), 21-26.

Hay, L. E., Wilby, R. L., & Leavesley, G. H. (2000). A comparison of delta change and downscaled GCM scenarios for three mountainous basins in the United States 1. *JAWRA Journal of the American Water Resources Association*, 36(2), 387-397.

Menne, M. J., Williams, C. N., Gleason, B. E., Rennie, J. J., & Lawrimore, J. H. (2018). The global historical climatology network monthly temperature dataset, version 4. *Journal of Climate*, 31(24), 9835-9854.

Fig. S13: Example to bias-correct this study's reported temperature offsets.

The paper also puts emphasis on the spatial structure (semivariograms) of open-air versus understory temperatures. The distances where spatial independence is reached (according to the exponential model used, which is a common choice) differ a lot between the two temperature variables. However, the numbers reported are not based on observational networks but on the ERA5 reanalysis data on one hand, which uses kriging for interpolation, and the RF outputs on the other, which uses explanatory variables e.g. related to topography. What we see here (in Fig. 4) is the spatial autocorrelation of the kriging procedure vs. that of the topography, not that of the temperature data per se. The qualitative result (shorter correlations for the understory) is intuitive, but the precise numbers reported (e.g. 44 km or 85 km) are questionable. The authors

seem to be aware that they don't quantify the spatial structure of temperature observations directly (l. 216ff), but should be more explicit on the issue and refrain from reporting these results at that precision (like "the understory temperature data in Central Amazonia Forest become spatially independent after 122 km").

Reply: We agree with this assessment, and we now only mention the **general** relationship between ERA5-Land temperature and modelled understory temperature and refrained from reporting precise numbers, as **suggested by** the reviewer (L209-214).

In that regard, the detrending performed prior to calculating semivariograms, based on best fit surfaces (l. 385ff and Fig. S4) is not described in sufficient detail. What are the horizontal axes in Fig. S4 (unreadable because of blur)? Why is using the residuals only essential? What type of bias would be induced when the detrending is skipped?

Reply: In a semivariogram analysis, it's important to identify and account for the overriding processes (global trends) (e.g., the change of temperature with elevation) because they can obscure the underlying spatial structure of the data. Once these trends are identified, for instance, using a linear model, they can be mathematically subtracted from the data, a process known as detrending. This allows for a more robust analysis of the spatial relationships.

In the revised manuscript, we've now incorporated additional information to indicate that a linear model was used to define the best fit surface for each dataset using the information of latitude, longitude, and temperature. This process of detrending was instrumental in addressing the dominant physical process that was evident in both datasets (most likely elevation) and predictably influenced the temperature values (L408-411). Fig. S4 is also updated with correct axes labelling (i.e., lat and long) and increased font size for better readability.

A positive highlight of the paper is the way it deals with interpolation vs. extrapolation (l. 420f). Where and how much the model is forced to extrapolate compared to the observations, is made rather explicit and precisely quantified based on a thorough PC analysis and 2D-convex hulls for 15 combinations of PC pairs. We have seen many papers where the information about the level of extrapolation is not revealed at all.

Reply: We appreciate the reviewer's acknowledgment of our efforts to present various facets of model uncertainties. We have indeed gone beyond reporting single-value accuracy metrics and have quantified spatially explicit model errors, to allow readers to understand the limitations of the study.

It would be tempting to try to verify the model predictions through observations of understory and open-air temperature in close vicinity. They might be scarce and scattered, but it should be possible to get access to 2 m air temperature time series from weather stations close to some of

your observations sites? This should be a much more stringent test instead of comparing model output with model output as done in the paper.

Reply: Considering the reviewer's suggestion, we conducted a comparative analysis between the monthly temperature values from weather stations and the modelled understory temperature in close proximity, across all seven countries where TMS sensors were installed. This comparison, depicted in Fig. S13 (c), underscores that the trained RF model can capture seasonal variations and aligns well with weather station observations.

Overall, while the data analysis methodology is by and large rather sophisticated and thorough, the observational basis with just a couple of months of records here and there is weak, and the paper tends to overstate its results. The limitations of the approach should be pointed out more clearly, following the comments given here.

Reply: We agree that data availability remains poor in the tropics, making it difficult to make accurate predictions across the whole region. Nevertheless, we believe that our rigorous analysis, and our open communication about the data's limitation, enable us to provide a nuanced story to the reader and with products that are highly beneficial for further applications. We try to emphasize this further throughout the manuscript by 1) recognizing the need for more ground observations to further improve the understory temperature mapping (L280-L282), 2) highlighting that this is a first attempt at modelling microclimate in a traditionally data-poor region (Table S5), 3) guiding the reader to our open data and methodological framework as available on online repository (L533-547), and 4) emphasizing the availability of uncertainty layers to allow users to mask regions of lower data quality in their applications (Fig. S9 , Fig. S12, Fig. S13).

Despite these limitations, we believe our results contribute to bridging a critical scientific gap by providing the first estimate of the spatial patterns of understory temperatures in the tropics. Our study goes beyond a static layer, providing a solid and transparent methodological framework that will be openly available. This framework will be used by the microclimate research community to interactively improve the results and reduce uncertainties.

Further, there are a non-negligible number of typos, some of them (by no means all) are corrected in the annotated pdf, which also contains further comments and suggestions for improvement. If considered thoroughly, the paper should have publication potential and serves the need for understory temperature data for understanding and modelling vegetation dynamics - not only in the tropics.

Reply: We appreciate the reviewer's recognition of the significance of this study and its potential for publication. Please note that we have addressed all the typographical errors highlighted in the PDF annotations in our revised manuscript.

REVIEWERS' COMMENTS

Reviewer #1 (Remarks to the Author):

This is a revision of a manuscript I previously reviewed favorably, but with some feedback and criticism. The authors have now addressed my comments and concerns, e.g., by providing links to output data and the code used to analyze the data. In addition, they conducted additional analyses and present additional (supplementary) results, based on comments by the other reviewer. I recognize that some of the issues raised by the other reviewer are quite critical. I am satisfied with the responses to my comments, and I believe the additional analyses to be valuable, but will leave it to the other reviewer to assess whether their comments have been adequately addressed with these additions.

248-253. I have a hard time believing this. Because the transpiration would have to happen in the understory for understory to be cooler than macroclimate. And why would this happen particularly at high elevation and not at low elevation?

289. need to be more specific. Elevation negative relationship with microclimate, as in, higher elevation lower T? or higher elevation, more difference between macro and microclimate?

L412. "shown" instead of "show"

Reviewer #2 (Remarks to the Author):

The revised version of this "understorey temperature" manuscript represents a significant improvement. The rebuttal letter provides a detailed account of changes made, and the impression that this is a thorough investigation into the deviations between below-canopy and open-air temperature data which is on one hand methodologically sound, and on the other very aware of its limitations induced by limited data availability for the local measurements (180 TMS loggers with records of a few months up to roughly two years) and biases for the remote-sensing based products used.

The reviewer has carefully checked the text changes made (difference between the original submission and this revised one) for both reviewers (also including #1) and doesn't see even minor compromises made. Very well done revision!

Here comes a completely minor set of typos remaining:

l. 177 is shown -> are shown

l. 201 RT is -> RT are

l. 258 absorbed -> absorbs

l. 412 show -> shown

l. 449 such situation -> such a situation

- I. 502 compared other -> compared to other
- I. 512 place "deg C" behind both RMSE and MAE
- I. 514 note that -> note is that
- I. 610 pixels -> pixel, value -> values, indicated values -> indicate that values
- I. 612 that model -> that the model

Conclusion: the paper can be accepted as is (please correct the typos)

Reviewer #1:

This is a revision of a manuscript I previously reviewed favorably, but with some feedback and criticism. The authors have now addressed my comments and concerns, e.g., by providing links to output data and the code used to analyze the data. In addition, they conducted additional analyses and present additional (supplementary) results, based on comments by the other reviewer. I recognize that some of the issues raised by the other reviewer are quite critical. I am satisfied with the responses to my comments, and I believe the additional analyses to be valuable, but will leave it to the other reviewer to assess whether their comments have been adequately addressed with these additions.

248-253. I have a hard time believing this. Because the transpiration would have to happen in the understory for understory to be cooler than macroclimate. And why would this happen particularly at high elevation and not at low elevation?

Reply: We do second the viewpoint of the reviewer, however, these arguments presented in L248-253 are largely based on published literature, and we decided to present these arguments to provide a comprehensive range of plausible explanations.

289. need to be more specific. Elevation negative relationship with microclimate, as in, higher elevation lower T? or higher elevation, more difference between macro and microclimate?

Reply: The Elevation negative relationship with microclimate (L289) means that temperature declines with increasing elevation – this is clearly demonstrated in Figure S6a. Perhaps the reviewer missed that the reference to Figure S6 is already mentioned in the previous sentence.

L412. “shown” instead of “show”

Reply: Please note we have corrected all the typos pointed out by the reviewer in the revised manuscript.

Reviewer #2:

The revised version of this "understorey temperature" manuscript represents a significant improvement. The rebuttal letter provides a detailed account of changes made, and the impression that this is a thorough investigation into the deviations between below-canopy and open-air temperature data which is on one hand methodologically sound, and on the other very aware of its limitations induced by limited data availability for the local measurements (180 TMS loggers with records of a few months up to roughly two years) and biases for the remote-sensing based products used.

The reviewer has carefully checked the text changes made (difference between the original submission and this revised one) for both reviewers (also including #1) and doesn't see even minor compromises made. Very well done revision!

Here comes a completely minor set of typos remaining:

l. 177 is shown -> are shown

l. 201 RT is -> RT are

l. 258 absorbed -> absorbs

l. 412 show -> shown

l. 449 such situation -> such a situation

l. 502 compared other -> compared to other

l. 512 place "deg C" behind both RMSE and MAE

l. 514 note that -> note is that

l. 610 pixels -> pixel, value -> values, indicated values -> indicate that values

l. 612 that model -> that the model

Conclusion: the paper can be accepted as is (please correct the typos)

Reply: Please note we have corrected all the typos pointed out by the reviewer in the revised manuscript.